# Childhood injury after a parental cancer diagnosis

**Ruoqing Chen[1]\*[†], Amanda Regodón Wallin[1][†], Arvid Sjölander[1], Unnur Valdimarsdóttir[1,2], Weimin Ye[1], Henning Tiemeier[3,4], Katja Fall[5], Catarina Almqvist[1,6], Kamila Czene[1], Fang Fang[1]**

[1]Department of Medical Epidemiology and Biostatistics, Karolinska Institutet, Stockholm, Sweden; [2]Center of Public Health Sciences, Faculty of Medicine, University of Iceland, Reykjavík, Iceland; [3]Department of Epidemiology, Erasmus MC University Medical Center, Rotterdam, The Netherlands; [4]Department of Child and Adolescent Psychiatry, Erasmus MC University Medical Center, Rotterdam, The Netherlands; [5]Clinical Epidemiology and Biostatistics, Faculty of Medicine and Health, Örebro University, Örebro, Sweden; [6]Lung and Allergy Unit, Astrid Lindgren Children's Hospital, Karolinska University Hospital, Stockholm, Sweden

**\*For correspondence:** ruoqing. chen@ki.se

[†]These authors contributed equally to this work

**Competing interests:** The authors declare that no competing interests exist.

**Abstract** A parental cancer diagnosis is psychologically straining for the whole family. We investigated whether a parental cancer diagnosis is associated with a higher-than-expected risk of injury among children by using a Swedish nationwide register-based cohort study. Compared to children without parental cancer, children with parental cancer had a higher rate of hospital contact for injury during the first year after parental cancer diagnosis (hazard ratio [HR] = 1.27, 95% confidence interval [CI] = 1.22-1.33), especially when the parent had a comorbid psychiatric disorder after cancer diagnosis (HR = 1.41, 95% CI = 1.08-1.85). The rate increment declined during the second and third year after parental cancer diagnosis (HR = 1.10, 95% CI = 1.07-1.14) and became null afterwards (HR = 1.01, 95% CI = 0.99-1.03). Children with parental cancer also had a higher rate of repeated injuries than the other children (HR = 1.13, 95% CI = 1.12-1.15). Given the high rate of injury among children in the general population, our findings may have important public health implications.

## Introduction

Cancer is not only a leading cause of morbidity and mortality among the affected patients, it is also increasingly recognized as a contributor to ill-health in their significant others (*Sjovall et al., 2009*; *Visser et al., 2004*; *Kazak et al., 2005*; *Krauel et al., 2012*). In Sweden, the number of newly diagnosed cancer patients has more than doubled during the last forty years and a considerable proportion of these patients are parenting minor children (*National Board of Health and Welfare, 2014*). A diagnosis of cancer in parents has repeatedly been shown to exert psychological and social stress in their children (*Visser et al., 2004*). Coping with cancer may affect the parenting of both the ill and well parents, further impacting the behavioral and social adaptability of the children (*Faulkner and Davey, 2002*). In contrast to the relatively rich literature on behavioral and mental well-being of children living with a parent with cancer, few studies have so far addressed somatic health outcomes among these children. In a recent study, we reported that children of parents with cancer had a higher risk of death, both due to cancer and other causes (*Chen et al., 2015*).

Injury is the most common cause of hospital care among children and accounts for almost one million child deaths annually worldwide (*Peden et al., 2008*). Sociodemographic, behavioral and psychosocial factors of both children and their family are known determinants of injuries among children

**eLife digest** A diagnosis of cancer can be devastating for both a person and his or her family. Over the past 40 years, the number of individuals in Sweden diagnosed with cancer has more than doubled leaving growing numbers of families coping with the aftermath. Many individuals diagnosed with cancer have young children. Parents with cancer and their spouses often struggle to cope with disease and the demands of parenting simultaneously. In fact, previous research has shown children with a parent who has cancer have a greater risk of behavioral problems or distress than children with two healthy parents.

Whether the stress of having a parent with cancer also affects the children's physical wellbeing hasn't been studied much. One concern in particular is whether these children may be at increased risk of injury. Injuries are the most common reason for a child to visit a hospital and in some cases lead to deaths. Children who are not well supervised or whose parents have poor mental health are at increased risk of injury. Coping with cancer and the mental anguish it causes may distract parents and possibly place their children at increased risk of injury.

Based on data from nationwide population and health registers in Sweden, Chen, Regodón Wallin et al. now provide evidence that a child with a parent who has cancer is at a greater risk of injury than a child with two parents who are free of cancer. The analysis also revealed that the risk is particularly great if the parent with cancer also develops mental illness after the cancer diagnosis. The risk of injury is greatest in the first year after the parent's diagnosis. Fortunately, the elevated risk of injury decreases overtime and is almost non-existing after the third year. The analyses suggest that providing extra support for parents with cancer might help to reduce the risk of injury in their children.

(*Horwitz et al., 1988*). For example, childhood injury has been associated with male sex, risk-taking behavior, lack of parental supervision as well as poor mental health of the parents (*Matheny, 1986*; *Schwebel et al., 2011*; *Morrongiello et al., 2006b*; *Peden et al., 2008*; *McKinlay et al., 2010*). To our knowledge, no study has however specifically addressed the impact of parental cancer diagnosis on the risk of child injury. To this end, we leveraged the nationwide population and health registers in Sweden to explore the association between parental cancer diagnosis and the risk of hospital contact for injury among children.

## Results

During the study period, 78,815 children (4%) were exposed to a parental cancer diagnosis. The general characteristics of the exposed children differed slightly from the unexposed children, in terms of larger number of siblings, shorter gestational age, higher proportion of delivery through caesarian section, higher proportions of birth weight <2500 or >4000 grams, higher proportion of maternal smoking in early pregnancy, higher paternal and maternal ages at child's birth, as well as higher educational level of the parents (*Table 1*).

### Primary analysis

During follow-up, 15,377 exposed children (incidence rate: 52 per 1000 person-years) and 548,488 unexposed children (incidence rate: 46 per 1000 person-years) had a first hospital contact for injury. Adjusting for only attained age and sex, the exposed children had a 4% higher rate of hospital contact for injury (hazard ratio [HR], 1.04; 95% confidence interval [CI], 1.02–1.05) than the other children. After adjustment for all covariates, the association became stronger (HR, 1.07 [95% CI 1.05–1.09]) (*Table 2*). Approximately 17% of hospital contacts among the exposed children occurred during the first year after cancer diagnosis, corresponding to an incidence rate of 60 per 1000 person-years and a HR of 1.27 (95% CI 1.22–1.33). The rate increment decreased during the second and third years, and became null after three years (*Table 2*).

The association was not modified by the sex of cancer parent or by the expected survival of cancer; the association did not differ between smoking/alcohol-related cancers and other cancers (*Table 2*; all p>0.05). However, children whose cancer parent had developed a comorbid psychiatric

**Table 1.** Characteristics of the participating children and their parents.

| Characteristics | All children N (%) | Children with parental cancer N (%) | Children without parental cancer N (%) | p |
|---|---|---|---|---|
| **Characteristics of the children** | | | | |
| **Sex** | | | | |
| Male | 1,008,982 (51.4) | 40,527 (51.4) | 968,455 (51.4) | 0.72 |
| Female | 955,645 (48.6) | 38,288 (48.6) | 917,357 (48.6) | |
| **No. of siblings and half siblings** | | | | |
| 0 | 189,556 (9.6) | 5,754 (7.3) | 183,802 (9.7) | <0.001 |
| 1 | 784,911 (40.0) | 28,901 (36.7) | 756,010 (40.1) | |
| 2 | 564,343 (28.7) | 23,291 (29.6) | 541,052 (28.7) | |
| ≥ 3 | 425,817 (21.7) | 20,869 (26.5) | 404,948 (21.5) | |
| **Gestational age (weeks)** | | | | |
| < 35 | 42,376 (2.2) | 1,833 (2.3) | 40,543 (2.1) | <0.001 |
| 35 - 36 | 70,999 (3.6) | 3,111 (3.9) | 67,888 (3.6) | |
| 37 - 38 | 363,508 (18.5) | 15,192 (19.3) | 348,316 (18.5) | |
| 39 - 40 | 973,949 (49.6) | 38,169 (48.4) | 935,780 (49.6) | |
| 41 - 42 | 456,222 (23.2) | 18,333 (23.3) | 437,889 (23.2) | |
| ≥ 43 | 12,898 (0.7) | 528 (0.7) | 12,370 (0.7) | |
| Missing | 44,675 (2.3) | 1,649 (2.1) | 43,026 (2.3) | |
| **Mode of delivery** | | | | |
| Caesarean section | 237,822 (12.1) | 10,300 (13.1) | 227,522 (12.1) | <0.001 |
| Vaginal delivery | 1,684,729 (85.8) | 66,971 (85.0) | 1,617,758 (85.8) | |
| Missing | 42,076 (2.1) | 1,544 (2.0) | 40,532 (2.1) | |
| **Birth weight (g)** | | | | |
| < 2500 | 78,412 (4.0) | 3,455 (4.4) | 74,957 (4.0) | <0.001 |
| 2500-2999 | 207,951 (10.6) | 8,361 (10.6) | 199,590 (10.6) | |
| 3000-3499 | 604,528 (30.8) | 23,466 (29.8) | 581,062 (30.8) | |
| 3500-3999 | 665,343 (33.9) | 26,714 (33.9) | 638,629 (33.9) | |
| 4000-4499 | 290,798 (14.8) | 12,039 (15.3) | 278,759 (14.8) | |
| ≥ 4500 | 68,620 (3.5) | 2,980 (3.8) | 65,640 (3.5) | |
| Missing | 48,975 (2.5) | 1,800 (2.3) | 47,175 (2.5) | |
| **Maternal smoking in early pregnancy** | | | | |
| No | 1,421,392 (72.4) | 55,388 (70.3) | 1,366,004 (72.4) | <0.001 |
| Yes | 383,760 (19.5) | 17,036 (21.6) | 366,724 (19.5) | |
| Missing | 159,475 (8.1) | 6,391 (8.1) | 153,084 (8.1) | |
| **Characteristics of the parents** | | | | |

*Table 1 continued on next page*

*Table 1 continued*

| Characteristics | All children N (%) | Children with parental cancer N (%) | Children without parental cancer N (%) | p |
|---|---|---|---|---|
| **Paternal age at child's birth (years)** | | | | |
| < 20 | 11,942 (0.6) | 166 (0.2) | 11,776 (0.6) | |
| 20-24 | 199,251 (10.1) | 4,232 (5.4) | 195,019 (10.3) | |
| 25-29 | 584,302 (29.7) | 16,318 (20.7) | 567,984 (30.1) | <0.001 |
| 30-34 | 628,352 (32.0) | 23,309 (29.6) | 605,043 (32.1) | |
| ≥ 35 | 540,780 (27.5) | 34,790 (44.1) | 505,990 (26.8) | |
| **Maternal age at child's birth (years)** | | | | |
| < 20 | 47,255 (2.4) | 840 (1.1) | 46,415 (2.5) | |
| 20-24 | 395,011 (20.1) | 9,283 (11.8) | 385,728 (20.5) | |
| 25-29 | 713,827 (36.3) | 23,318 (29.6) | 690,509 (36.6) | <0.001 |
| 30-34 | 547,990 (27.9) | 25,787 (32.7) | 522,203 (27.7) | |
| ≥ 35 | 260,544 (13.3) | 19,587 (24.9) | 240,957 (12.8) | |
| **Highest educational level** | | | | |
| Primary school or lower | 98,230 (5.0) | 4,144 (5.3) | 94,086 (5.0) | |
| Secondary education | 987,431 (50.3) | 36,105 (45.8) | 951,326 (50.5) | |
| Tertiary education | 844,523 (43.0) | 36,812 (46.7) | 807,711 (42.8) | <0.001 |
| Postgraduate education | 34,063 (1.7) | 1,748 (2.2) | 32,315 (1.7) | |
| Missing | 380 (0.0) | 6 (0.0) | 374 (0.0) | |

**Table 2.** Hazard ratios for hospital contact for injury among children with parental cancer compared to children without parental cancer.

| Characteristics | Any Time After Parental Cancer Diagnosis | | | | First Year After Parental Cancer Diagnosis | | | |
|---|---|---|---|---|---|---|---|---|
| | No. of Children With a Hospital Contact for Injury | Person-years | HR (95% CI) * | p (Wald Test) | No. of Children With a Hospital Contact for Injury | Person-years | HR (95% CI) * | p (Wald Test) |
| No parental cancer | 548,488 | 11,879,075 | 1 | | 548,488 | 11,879,075 | 1 | |
| Parental cancer | 15,377 | 298,302 | 1.07 (1.05-1.09) | | 2,674 | 44,600 | 1.27 (1.22-1.33) | |
| Time since cancer diagnosis | | | | | | | | |
| ≤ 1 year | 2,674 | 44,600 | 1.27 (1.22-1.33) | | — | — | — | |
| >1 and ≤3 years | 3,850 | 74,087 | 1.10 (1.07-1.14) | <0.001 | — | — | — | |
| > 3 years | 8,853 | 179,615 | 1.01 (0.99-1.03) | | — | — | — | |
| Sex of the cancer parent | | | | | | | | |
| Male | 6,554 | 126,277 | 1.08 (1.05-1.11) | 0.48 | 1,166 | 18,917 | 1.32 (1.24-1.40) | 0.13 |
| Female | 8,823 | 172,026 | 1.06 (1.04-1.09) | | 1,508 | 25,683 | 1.24 (1.18-1.31) | |
| Tobacco-related cancer † | | | | | | | | |
| No | 12,008 | 233,848 | 1.07 (1.05-1.09) | 0.72 | 2,142 | 35,080 | 1.29 (1.24-1.35) | 0.13 |
| Yes | 3,369 | 64,454 | 1.08 (1.04-1.12) | | 532 | 9,520 | 1.20 (1.10-1.31) | |
| Alcohol-related cancer ‡ | | | | | | | | |
| No | 10,464 | 201,389 | 1.08 (1.05-1.10) | 0.30 | 1,745 | 28,525 | 1.30 (1.24-1.37) | 0.16 |
| Yes | 4,913 | 96,913 | 1.06 (1.02-1.09) | | 929 | 16,076 | 1.23 (1.15-1.31) | |
| Predicted 5-year relative survival rate | | | | | | | | |
| < 20% § | 931 | 18,845 | 1.02 (0.95-1.10) | 0.21 | 160 | 3,041 | 1.15 (0.98-1.35) | 0.38 |
| 20-80% | 7,112 | 136,080 | 1.08 (1.06-1.11) | | 1,243 | 20,736 | 1.27 (1.19-1.35) | |
| ≥ 80% ¶ | 7,334 | 143,377 | 1.06 (1.04-1.09) | | 1,271 | 20,824 | 1.30 (1.23-1.38) | |

*Table 2 continued on next page*

*Table 2 continued*

| Characteristics | Any Time After Parental Cancer Diagnosis | | | | First Year After Parental Cancer Diagnosis | | | |
|---|---|---|---|---|---|---|---|---|
| | No. of Children With a Hospital Contact for Injury | Person-years | HR (95% CI) * | p (Wald Test) | No. of Children With a Hospital Contact for Injury | Person-years | HR (95% CI) * | p (Wald Test) |
| **Parental psychiatric comorbidity after cancer diagnosis** ‖ | | | | | | | | |
| No | 14,630 | 285,621 | 1.06 (1.05-1.08) | 0.001 | 2,611 | 43,663 | 1.27 (1.22-1.32) | 0.45 |
| Yes | 747 | 12,681 | 1.21 (1.12-1.31) | | 63 | 938 | 1.41 (1.08-1.85) | |

HR, hazard ratio; CI, confidence interval

* Adjusted for attained age, sex, number of siblings, gestational age, mode of delivery and birth weight of the child, paternal age at child's birth, maternal age at child's birth, maternal smoking during early pregnancy, and the highest educational level of the parents.

† Tobacco-related cancers include cancers in lung, oesophagus, larynx, pharynx, mouth, lip, salivary glands, tongue, stomach, urinary bladder, kidney, uterine cervix, colon and pancreas.

‡ Alcohol-related cancers include cancers in liver, oral cavity, pharynx, larynx, oesophagus, colorectum and breast.

§ Including cancers in esophagus, liver, gall bladder, biliary tract, pancreas, lung and stomach.

¶ Including cancers in lip, breast, corpus uteri, testis, skin, thyroid and other endocrine glands, and Hodgkin's lymphoma.

‖ Including depression, anxiety disorders, stress reaction and adjustment disorder.

disorder after diagnosis had a higher rate of childhood injury (HR, 1.21 [95% CI 1.12–1.31], compared with children whose cancer parent had no such disease (HR, 1.06 [95% CI 1.05–1.08]) (p = 0.001). As in the overall analysis, the rate increment in these analyses was more prominent during the first year after diagnosis (*Table 2*).

The overall association was significantly stronger for boys than for girls (p for interaction, < 0.001) (*Table 3*). When focusing on the first year following parental cancer, no statistically significant difference was however detected between boys and girls (p = 0.17). Neither the overall association nor the association during the first year after parental cancer was modified otherwise by age at follow-up or number of siblings of the child (*Table 3*).

Among all hospital contacts, 96% were due to unintentional injuries (HR, 1.07 [95% CI 1.05–1.09]). Parental cancer also tended to be associated with a higher rate of intentional self-harm (HR, 1.09 [95% CI 0.95–1.25]) and undetermined or other injuries (HR, 1.11 [95% CI 0.98–1.26]), but not of assault (HR, 0.99 [95% CI 0.87–1.13]). The associations did not appear to further differ by nature, body region, or mechanism of injury, or by place of injury occurrence, either during the entire follow-up or during the first year after cancer diagnosis (*Figure 1*).

Among all events of injury, outpatient visit and hospitalization accounted for 83.5% and 16.5% respectively. Although the positive association was only statistically significant for outpatient visit during the entire follow-up (outpatient visit HR, 1.08 [95% CI 1.06–1.10]; hospitalization HR, 1.03 [95% CI 0.99–1.08]), the association was statistically significant for both hospitalization (HR, 1.18 [95% CI 1.07–1.31]) and outpatient visit (HR, 1.29 [95% CI 1.24–1.35]) during the first year after cancer diagnosis.

## Secondary analysis

With 7-day washout periods, the mean number of hospital contacts for injury was 1.8 during the study period. Among children with one previous hospital contact, parental cancer was associated with a 1.24-fold rate of having a second hospital contact (95% CI 1.20–1.28) (*Table 4*). Similar patterns were observed for children with 2–4 previous injuries (*Table 4*). When all injuries were studied, children with parental cancer had a 13% higher rate of repeated injuries (HR, 1.13 [95% CI 1.12–1.15]). Additional analyses with 14-day and 30-day washout periods showed similar results (14-day HR, 1.12 [95% CI 1.10–1.13]; 30-day HR, 1.10 [95% CI 1.08–1.12]).

## Discussion

In this nationwide register-based study, we found that children having a parent with cancer had a higher rate of hospital contact for injury compared with other children. The rate increment was noted for children of all ages as well as for different kinds of injuries or places of injury occurrence, but was most pronounced immediately after the parent's cancer diagnosis and among children with previous injuries. Comorbid psychiatric diagnoses after the cancer diagnosis rendered further higher rate increment of childhood injury.

Although it has been suggested that adolescents are most prone to psychosocial problems at the time of stressful life experience, younger children are in greater need of supervision and parenting (*Phillips, 2014*; *Macpherson and Emeleus, 2007a*; *MacPherson and Emeleus, 2007b*). The positive association between parental cancer and hospital contact for injury among children at all ages in the present study may therefore be jointly attributable to both the psychological distress among the children and the potential lack of parental supervision needed for injury prevention (*Morrongiello et al., 2006a*; *Davis Kirsch et al., 2003*; *Faulkner and Davey, 2002*; *Asbridge et al., 2014*; *Bylund Grenklo et al., 2013*). It has been debated whether boys and girls are differently affected by parental cancer (*Krattenmacher et al., 2012*; *Visser et al., 2005*). Our findings show that overall boys had a more pronounced rate increment than girls for injury. At the same age, boys are on average less mature in terms of social-emotional functioning compared to girls (*Visser et al., 2005*). However, worth noting is that despite the overall difference, boys and girls had similarly increased rates of injury, during the first year after parental cancer diagnosis.

Parental cancer was associated with a higher rate of injury, regardless of nature, mechanism, body region of the injury, or place of injury occurrence. Although a positive association was mainly noted for unintentional injuries, the lack of statistical significance for intentional injuries might be due to the relatively small number of intentional injuries observed. Interestingly and reassuringly, we

**Table 3.** Hazard ratios for hospital contact for injury among children with parental cancer compared to children without parental cancer, according to sex, age and number of full and half siblings of the child.

| Characteristics of the Child | No Parental Cancer | | | Any Time After Parental Cancer Diagnosis | | | | First Year After Parental Cancer Diagnosis | | | |
|---|---|---|---|---|---|---|---|---|---|---|---|
| | No. of Children With a Hospital Contact for Injury | Person-years | HR (95% CI) | No. of Children With a Hospital Contact for Injury | Person-years | HR (95% CI) | p for interaction | No. of Children With a Hospital Contact for Injury | Person-years | HR (95% CI) | p for interaction |
| **Sex*** | | | | | | | | | | | |
| Male | 313,806 | 5,966,451 | 1 | 9,088 | 150,070 | 1.11 (1.08-1.13) | < 0.001 | 1,594 | 22,747 | 1.30 (1.24-1.37) | 0.17 |
| Female | 234,682 | 5,912,624 | 1 | 6,289 | 148,233 | 1.02 (0.99-1.05) | | 1,080 | 21,854 | 1.23 (1.16-1.31) | |
| **Age (years)†** | | | | | | | | | | | |
| < 3 | 35,157 | 876,761 | 1 | 103 | 2,106 | 1.21 (0.99-1.47) | < 0.001 | 57 | 1,155 | 1.25 (0.96-1.63) | 0.72 |
| 3-5 | 55,452 | 1,508,188 | 1 | 439 | 11,775 | 1.07 (0.97-1.18) | | 131 | 3,181 | 1.19 (1.00-1.42)§ | |
| 6-11 | 197,984 | 4,625,786 | 1 | 4,085 | 88,106 | 1.08 (1.05-1.12) | 0.60 | 756 | 14,435 | 1.24 (1.15-1.34) | |
| 12-15 | 134,995 | 2,500,216 | 1 | 4,698 | 83,199 | 1.08 (1.04-1.11) | | 777 | 11,429 | 1.27 (1.18-1.37) | |
| ≥ 15 | 124,900 | 2,368,124 | 1 | 6,052 | 113,116 | 1.06 (1.03-1.09) | | 953 | 14,400 | 1.32 (1.23-1.41) | |
| **No. of full and half siblings‡** | | | | | | | | | | | |
| 0 | 56,174 | 1,346,435 | 1 | 1,143 | 24,301 | 1.05 (0.98-1.11) | 0.13 | 208 | 3,759 | 1.30 (1.13-1.50) | 0.74 |
| 1 | 229,145 | 4,982,310 | 1 | 5,806 | 113,772 | 1.06 (1.04-1.09) | | 987 | 16,907 | 1.24 (1.16-1.32) | |
| 2 | 151,985 | 3,242,395 | 1 | 4,356 | 85,386 | 1.05 (1.02-1.09) | | 782 | 12,748 | 1.28 (1.19-1.38) | |
| ≥ 3 | 111,184 | 2,307,935 | 1 | 4,072 | 74,843 | 1.11 (1.07-1.15) | | 697 | 11,187 | 1.31 (1.21-1.41) | |

HR, hazard ratio; CI, confidence interval

*Adjusted for attained age and sex of the child, interaction between sex of the child and cancer of the parents, number of siblings, gestational age, mode of delivery and birth weight of the child, paternal age at child's birth, maternal age at child's birth, maternal smoking during early pregnancy, and the highest educational level of the parents.

[†]Adjusted for attained age of the child, interaction between attained age of the child and cancer of the parents, sex, number of siblings, gestational age, mode of delivery and birth weight of the child, paternal age at child's birth, maternal age at child's birth, maternal smoking during early pregnancy, and the highest educational level of the parents.

[‡]Adjusted for attained age, sex and number of siblings of the child, interaction between number of siblings of the child and cancer of the parents, gestational age, mode of delivery and birth weight of the child, paternal age at child's birth, maternal age at child's birth, maternal smoking during early pregnancy, and the highest educational level of the parents

[§]p = 0.054

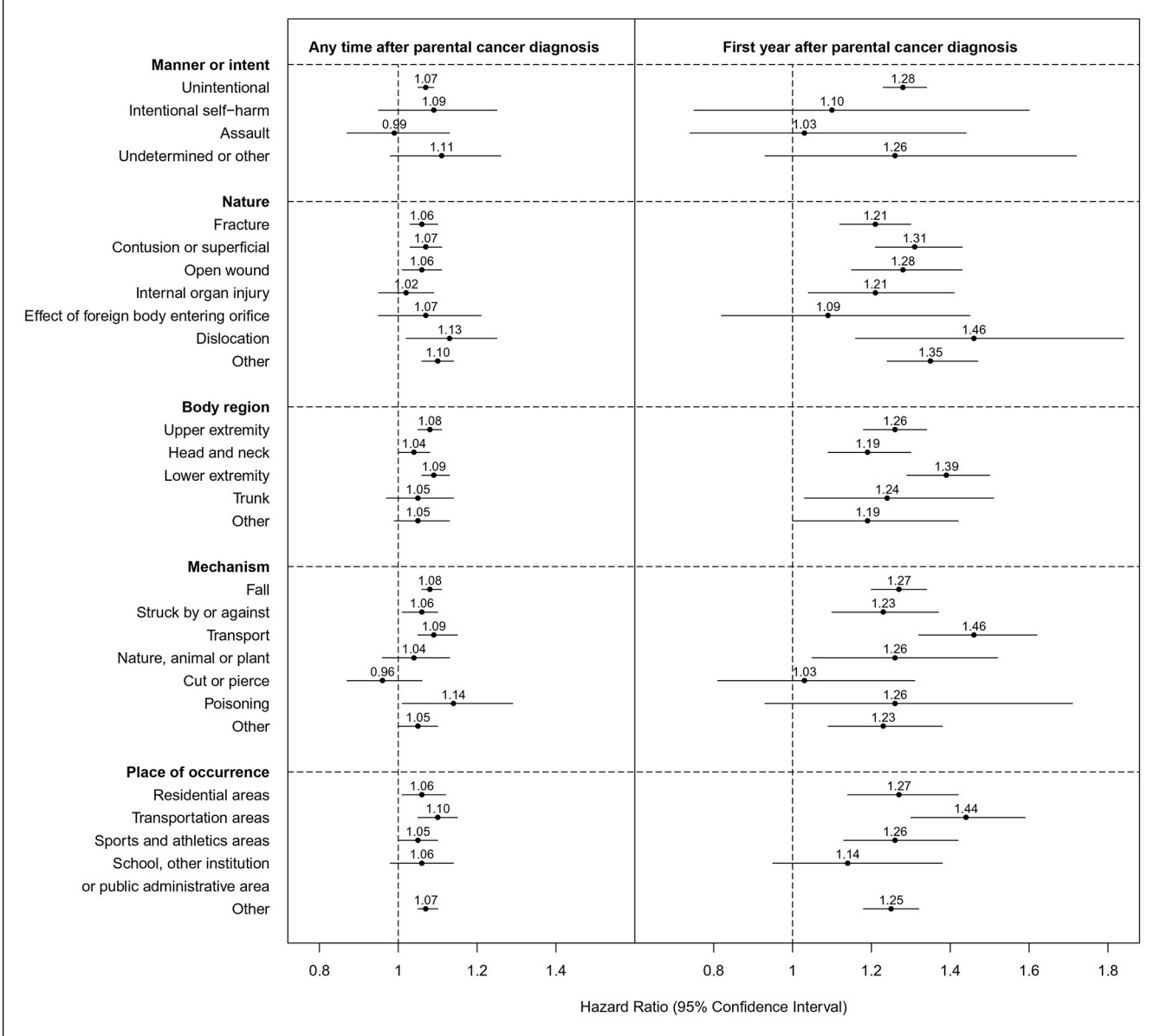

**Figure 1.** Hazard ratios for hospital contacts for injury among children with parental cancer compared to children without parental cancer, according to different characteristics of injury (Hazard ratios were adjusted for attained age, sex, number of siblings, gestational age, mode of delivery and birth weight of the child, paternal age at child's birth, maternal age at child's birth, maternal smoking during early pregnancy, and the highest educational level of the parents).

found no increased rate of assault-related injuries after parental cancer. The fact that the higher rate of injury was noted not only at home, but also in transportation areas, in sports areas, etc., suggests that efforts in preventing injuries in children living with a parent with cancer should include a larger circle of support.

Children's adjustment appears to vary at different stages of their parent's cancer disease (*Nelson and While, 2002*; *Huizinga et al., 2010*). Our results showed clearly that children had the highest injury rate increase during the first year after the parent's cancer diagnosis. This finding corroborates earlier findings in indicating that a cancer diagnosis poses severe psychological distress immediately after the diagnosis, both among the cancer patients and among their children (*Fang et al., 2012*; *Lu et al., 2013*; *Fall et al., 2009*; *Huizinga et al., 2010*). Previous studies have demonstrated that the well-being of children living with a parent with cancer is largely dependent

**Table 4.** Hazard ratios for hospital contact for injury among children with parental cancer compared to children without parental cancer, according to the number of previous hospital contact for injury of the child

| Characteristics | No. of Children With a Hospital Contact for Injury | Person-years | HR (95%CI) * |
| --- | --- | --- | --- |
| No contact | | | |
| No parental cancer | 548,488 | 11,879,075 | 1 |
| Parental cancer | 15,377 | 298,302 | 1.07 (1.05-1.09) |
| One contact | | | |
| No parental cancer | 228,560 | 1,542,926 | 1 |
| Parental cancer | 5,981 | 31,308 | 1.24 (1.20-1.28) |
| Two contacts | | | |
| No parental cancer | 107,833 | 454,049 | 1 |
| Parental cancer | 2,796 | 9,130 | 1.26 (1.20-1.32) |
| Three contacts | | | |
| No parental cancer | 53,784 | 176,237 | 1 |
| Parental cancer | 1,377 | 3,691 | 1.21 (1.13-1.31) |
| Four contacts | | | |
| No parental cancer | 28,126 | 74,499 | 1 |
| Parental cancer | 722 | 1,600 | 1.19 (1.07-1.33) |

HR, hazard ratio; CI, confidence interval

*Adjusted for attained age, sex, number of siblings, gestational age, mode of delivery and birth weight of the child, paternal age at child's birth, maternal age at child's birth, maternal smoking during early pregnancy, and the highest educational level of the parents

on the adjustment status of their parents to the cancer (*Krattenmacher et al., 2012*; *Nelson and While, 2002*; *Thastum et al., 2009*; *Huizinga et al., 2011*). This was supported in our findings that children whose parent was also diagnosed with a psychiatric disorder after cancer diagnosis, appeared to have more pronounced rate increase of injury. Furthermore, among children with higher baseline risk of injury (i.e., children with previous hospital contact for injury), parental cancer was associated with an even more elevated risk for future injuries. These results highlight both a high-risk time window and high-risk groups for potential future interventions. Although the number of children living with a parent of cancer will undoubtedly increase due to the increasing cancer incidence and improving cancer survival, the postponement of childbearing, etc., the proportion of such children is however still small, making dedicated intervention both feasible and viable.

Conflicting results have been reported regarding whether maternal cancer has greater adverse impact on children than paternal cancer (*Visser et al., 2005*; *Compas et al., 1994*). In line with our previous finding on child mortality after parental cancer, the present study indicated no difference between maternal and paternal cancer in relation to the consequent risk of childhood injury (*Chen et al., 2015*). In contrast to previous findings, we found no difference in child injury risk by the severity of parental cancer (*Krattenmacher et al., 2012*). One potential explanation may be the fact that the severity of cancer does not always positively correlate with the adjustment status of the cancer patients. For example, in a recent large-scale study, it was reported that patients of cancers of relatively better survival (e.g., breast cancer) had the highest prevalence of mental disorders, whereas patients of cancers with much severe prognosis (e.g., pancreatic cancer) had the lowest prevalence (*Mehnert et al., 2014*).

This study is the first to use a population-based sample to examine the impact of parental cancer on the risk of childhood injury. The major strengths of our study include the nationwide cohort design, using the effective record linkage across the high quality Swedish population and health registers and the prospectively and independently collected information on exposure and outcome. These strengths enhance clearly the validity and generalizability of our findings. Some limitations of our study also deserve consideration. For instance, we had no information on the cohabitation or employment status of the parents. A cancer diagnosis may have considerable impact on the marital relationship and the family's economic status (*Wozniak and Izycki, 2014*; *de Boer et al., 2009*)

which may in turn trigger additional psychological distress of the parents, leading to suboptimal parenting (*Tein et al., 2000*; *Sallinen et al., 2004*). Divorce or separation may further contribute to child injuries due to the departure of one parent from the household or simply a joint custody of the child between the parents. Therefore potential modifying effect of residence status within the family, as well as the cohabitation and employment status of the parents on the studied association deserves further investigation. Residual confounding due to unmeasured or unknown confounders is possible, however, with presumably small impact. In the multivariable models, only adjustment for the age of the child and the parental ages at child's birth had noticeable impact on the increase in injury risk, whereas adjustment for other covariates including birth characteristics of the child and educational level of the parents, which have previously been suggested to be associated with both injury risk among the children and cancer risk among the parents, had rather negligible impact (data not shown) (*Innes and Byers, 2004*; *Sun et al., 2010*; *Davey Smith et al., 2007*; *Beiki et al., 2014*; *Hemminki and Li, 2003*). The facts that the rate increment was mainly noted during the first three years after parental cancer diagnosis but not thereafter, and that the rate increment was independent of number of siblings or whether the cancer is smoking/alcohol-related or not, further alleviated concerns about residual confounding. Misclassification of injuries remains possible as only above 80% of outpatient visits were included in the Patient Register currently (*National Board of Health and Welfare, 2009*). However, such misclassification is largely administrative and arguably non-differential. Cancer parents with established contact with health care may be more likely to seek medical care for their children's injury. Yet, the opposite can also be postulated that while coping with this major illness, parents are less likely to seek medical care for minor hassles of their children. Such misclassification, if it exists, should have little impact on inpatient care for injury – a proxy of more severe injury event – and could not explain the largely increased injury rate during the first three years after parental cancer diagnosis whereas not thereafter.

In summary, children with a parent of cancer had a greater rate of hospital contact for injury, especially during the first year after cancer diagnosis. The association was also more pronounced for parental cancer with comorbid psychiatric disorders after the cancer diagnosis and among children with previous injuries.

## Materials and methods

### Study participants

We conducted a historical cohort study from 2001 to 2010, including all children born in Sweden during 1983–2002 (n = 2,071,380) based on the Swedish Multi-Generation Register. The Swedish Multi-Generation Register contains information on all residents in Sweden who were born from 1932 onward and alive in 1961, together with their parents (*Statistics Sweden, 2011*). To be included in the present study, a child must have both biological parents alive, free of cancer and identifiable from this register before the child's birth (n = 2,027,863).

### Parental cancer

All parents of these children were linked to the Swedish Cancer Register, which contains almost 100% complete information on all newly diagnosed cancer cases in Sweden since 1958 (*Barlow et al., 2009*). Information on type of cancer and date of diagnosis was collected from this register. If both parents were diagnosed with a cancer, the first diagnosis was used.

### Childhood injury

A hospital contact for injury was identified as either a hospitalization or an outpatient visit with injury according to the Swedish Patient Register. This register was initiated in 1964/1965 and has national coverage for hospital discharge records since 1987 (*Ludvigsson et al., 2011*). Since 2001, it also collects information on hospital-based outpatient specialist visits with over 80% coverage of the entire country (*Ludvigsson et al., 2011*). Information collected includes dates of admission and discharge, primary as well as multiple secondary diagnoses, and additionally external causes of morbidity and mortality when applicable. All diagnoses and external causes are coded according to Swedish revisions of the International Classification of Diseases (ICD). Since we were primarily interested in non-medical injuries, injuries due to complications of medical and surgical care were excluded from the

definition of childhood injury in the present study. Thus, to be defined as a hospital contact for injury, the record had to have a main discharge diagnosis of injury (ICD 10: S00-T98 except T80-T88, T98.3) and an external cause (ICD 10: V01-Y98 except Y40-Y84, Y88).

In the primary analysis, we used the first hospital contact for injury during follow-up as the outcome and the date of admission or outpatient visit as the date of injury occurrence. To further examine whether the impact of parental cancer diagnosis differed between any hospital contact for injury (i.e., first hospital contact) and repeated hospital contacts for injury, we analyzed children that had more than one hospital contact for injury during the study period. In this secondary analysis, all hospital visits within a 7-day time period (wash-out period) was counted as one contact (i.e., more likely referring to the same injury). In additional analyses, we also used 14-day and 30-day wash-out periods to assess the robustness of this definition.

## Follow-up

In the primary analysis, all children were followed from January 1, 2001 or date of birth, whichever came later. Children without parental cancer contributed person-time to the unexposed period, whereas children with parental cancer contributed person-time first to the unexposed period and after date of parental cancer diagnosis to the exposed period. Children who had a parent diagnosed with cancer before January 1, 2001 contributed all person–time to the exposed period. For both exposed and unexposed periods, the follow-up was censored on the date of first hospital contact for injury, emigration, death, 18th birthday, or December 31, 2010, whichever occurred first. As a result, 63,236 children who had died or emigrated or became 18 years old before/at the start of follow-up were excluded, leaving 1,964,627 children in the final analyses.

In the secondary analysis, we specifically followed children (both exposed and unexposed) who already had one hospital contact for injury, from the end of wash-out period to the following injuries. For example, to examine the association of parental cancer diagnosis and a future injury among children that had already one hospital visit for injury, we followed all children with a first hospital contact for injury to the second one. Similar follow-ups were conducted when examining the risk of a third, fourth, etc. hospital contact for injury.

## Covariates

Various characteristics in children and parents have been linked to both risks of child injury and parental cancer and therefore might either confound or modify the studied association (*Boutsikou and Malamitsi-Puchner, 2011*; *Morrongiello et al., 2007*; *Bradbury et al., 1999*; *Peden et al., 2008*; *Innes and Byers, 2004*; *Hjern, 2012*; *Weitzman et al., 1992*; *Weitoft et al., 2003*; *Sun et al., 2010*; *Hemminki and Li, 2003*). To address potential confounding and effect modification, we collected information on sex, gestational age, mode of delivery and birth weight of the child, maternal smoking during early pregnancy and maternal age at child's birth from the Swedish Medical Birth Register, as well as number of full and half siblings of the child and paternal age at child's birth through the Multi-Generation Register. The Medical Birth Register was established in 1973 and has covered over 99% of all births in Sweden since 1983 (*Centre for Epidemiology, National Board of Health and Welfare, 2003*). We further identified the highest educational level of the parents from the Swedish Register of Education (Statistics Sweden. 2004).

## Data availability

The summary of data included in different registers used in the present study can be found on the homepages of the Swedish National Board of Health and Welfare (http://www.socialstyrelsen.se/register) as well as the Statistics Sweden (http://www.scb.se/sv_/Vara-tjanster/bestalla-mikrodata/Vilka-mikrodata-finns/).

The authors confirm that, for approved reasons, some access restrictions apply to the data underlying the findings. The data used in this study are owned by the Swedish National Board of Health and Welfare and Statistics Sweden. According to Swedish law, the authors are not able to make the dataset publicly available.

Any researchers (including international researchers) interested in obtaining the data can do so by the following steps: 1) apply for ethical approval from their local ethical review boards; 2) contact the Swedish National Board of Health and Welfare and/or Statistics Sweden with the ethical approval

and make a formal application of use of register data. Contact emails for request of register data: Swedish National Board of Health and Welfare: registerservice@socialstyrelsen.se, Statistics Sweden: Mikrodata.individ@scb.se.

Please visit http://www.socialstyrelsen.se/register/bestalladatastatistik/bestallaindividuppgifterfor-forskningsandamal (the Swedish National Board of Health and Welfare) and http://www.scb.se/sv_/ Vara-tjanster/bestalla-mikrodata/ (the Statistics Sweden) for detailed information about how to apply for access to register data for research purposes.

## Statistical analysis

Pearson's $\chi^2$ test was used to compare the distributions of different child's and parental characteristics between the exposed and unexposed children.

## Primary analysis

Cox proportional hazards regression was used to compare the rate of first hospital contact for injury between children with and without parental cancer. HR with 95% CI was estimated after adjustment for the covariates described above. To account for the correlation among children of the same parents, we used "clustered" (sandwich) standard errors in all models. Time since birth was used as the underlying time scale in the Cox models; no statistically significant violation of the proportional hazards assumption was detected from a test of the Schoenfeld residuals. Parental cancer diagnosis was treated as a time-varying exposure.

To examine the specific impact of cancer diagnosis, independent of the later course of the disease, we calculated the HRs of first hospital contact for injury during the first year, >1 and ≤3 years, and >3 years after parental cancer diagnosis separately. Children with a parental cancer diagnosed before start of follow-up might not contribute to all three categories, depending on when the parental cancer was diagnosed and when follow-up was censored. Since we used time since birth as the underlying time scale, different HRs observed from these analyses did not conflict with the proportional hazards assumption tested. We sub-grouped parental cancer to explore whether maternal and paternal cancer had a different impact on child injury. To assess the impact of lifestyle factors as potential confounders for the studied association, we sub-grouped parental cancer as tobacco-related and other cancers, or alcohol-related and other cancers (*National Board of Health and Welfare, 2013*; World Health Organization). To explore the potential modifying effect of cancer severity, we further categorized parental cancer as cancer with high, medium or low expected 5-year survival. The expected 5-year survival was indexed as the predicted 5-year relative survival rates of different cancer types based on the entire Cancer Register (*Talback et al., 2004*). We further ascertained from the Patient Register hospital contacts for selected psychiatric comorbidity that were newly diagnosed after the cancer diagnosis among the parents. The psychiatric diagnoses considered were depression, anxiety disorders, stress reaction and adjustment disorder (detailed diagnoses and corresponding ICD codes are listed in the *Table 5*). We performed Wald tests to compare the HRs for different subgroups.

To assess whether the impact of parental cancer on child injury differed by sex, age or number of siblings of the child, we used formal tests of interaction of parental cancer with sex, age at follow-up (<3, 3–5, 6–11, 12–15 or ≥ 15 years), or number of full and half siblings (0, 1, 2, ≥3) of the child.

To examine whether the studied association differed for different types of injury, we further conducted separate analyses by manner or intent, nature, body region and mechanism of injury, as well as by place of injury occurrence. To assess whether the association varied by different severity of injury, we also examined separately the risk of hospitalization and outpatient visit for injury.

**Table 5.** Swedish revisions of the international classification of diseases (ICD) for psychiatric comorbidity of the cancer parents.

|  | ICD 8 (1969-1986) | ICD 9 (1987-1996) | ICD-10 (1997-presesnt) |
|---|---|---|---|
| Depression | 296.2, 298.0, 300.4 | 296B, 300E, 311 | F32-F39 |
| Anxiety disorders | 300 except 300.3, 300.4 | 300 except 300D, 300E | F40, F41, F44, F45, F48 |
| Stress reaction and adjustment disorder | 307 | 308, 309 | F43 |

### Secondary analysis

Ordinary Cox proportional hazard regression was used to assess the association between parental cancer diagnosis and a new injury among children with at least one previous hospital contact for injury during follow-up. A conditional Cox model (PWP-TT model) was used to assess the overall association between parental cancer diagnosis and repeated injuries in children (*Amorim and Cai, 2015*).

For all analyses, statistical significance was assessed using 2-tailed 0.05-level tests. Data preparation was performed using SAS version 9.4. Statistical analyses were performed using Stata version 12.1.

The study was approved by the Central Ethical Review Board in Stockholm, Sweden. All individuals' information was anonymized and de-identified prior to analysis.

## Acknowledgements

The authors wish to thank Dr Eva Norén Selinus for helpful conversations regarding diagnosis of psychiatric disorders in patients with cancer.

## Additional information

### Funding

| Funder | Grant reference number | Author |
|---|---|---|
| Forskningsrådet för Hälsa, Arbetsliv och Välfärd | 2012-0498 | Fang Fang |
| China Scholarship Council | 201206100002 | Ruoqing Chen |
| Svenska Sällskapet för Medicinsk Forskning | Researcher position | Fang Fang |
| Karolinska Institutet | Funding for Strategic Young Scholar Grants in Epidemiology | Catarina Almqvist Fang Fang |
| Karolinska Institutet | Assistant professor position | Fang Fang |
| Vetenskapsrådet | SIMSAM 340-2013-5867 | Weimin Ye Catarina Almqvist |
| Vetenskapsrådet | SIMSAM 80748301 | Weimin Ye Catarina Almqvist |

The funders had no role in study design, data collection and interpretation, or the decision to submit the work for publication.

### Author contributions

RC, FF, Obtained funding, Conception and design, Analysis and interpretation of data, Drafting or revising the article; ARW, AS, Conception and design, Analysis and interpretation of data, Drafting or revising the article; UV, HT, KF, Conception and design, Drafting or revising the article; WY, CA, Obtained funding, Conception and design, Acquisition of data, Drafting or revising the article; KC, Conception and design, Acquisition of data, Drafting or revising the article

### Author ORCIDs

Ruoqing Chen, http://orcid.org/0000-0003-4911-3543
Henning Tiemeier, http://orcid.org/0000-0002-4395-1397

### Ethics

Human subjects: The study was approved by the Central Ethical Review Board (Centrala etikprövningsnämnden) in Stockholm, Sweden (Dnr Ö 12-2013). In accordance with their decision, we did not obtain informed consent from participants involved in the study. All individuals' information was anonymized and de-identified prior to analysis.

# Additional files

## Supplementary files

• Reporting standards: Standard used to collect data: The Strengthening the Reporting of Observational Studies in Epidemiology (STROBE) Statement: guidelines for reporting observational studies

## Major datasets

The following dataset was generated:

| Author(s) | Year | Dataset title | Dataset URL | Database, license, and accessibility information |
|---|---|---|---|---|
| Almqvist C, Ye W, Czene K | 2015 | Swedish national registers | http://www.socialstyrelsen.se/register; http://www.scb.se/sv_/Vara-tjanster/bestalla-mikrodata/Vilka-mikrodata-finns/ | The authors confirm that, for approved reasons, some access restrictions apply to the data underlying the findings. The data used in this study are owned by the Swedish National Board of Health and Welfare and Statistics Sweden. According to Swedish law, the authors are not able to make the dataset publicly available. Any researchers (including international researchers) interested in obtaining the data can do so by the following steps: 1) apply for ethical approval from their local ethical review boards; 2) contact the Swedish National Board of Health and Welfare and/or Statistics Sweden with the ethical approval and make a formal application of use of register data. Contact emails for request of register data: Swedish National Board of Health and Welfare: registerservice@socialstyrelsen.se, Statistics Sweden: Mikrodata.individ@scb.se. Please visit http://www.socialstyrelsen.se/register/bestalladatastatistik/bestallaindividuppgifter-forforskningsandamal (the Swedish National Board of Health and Welfare) and http://www.scb.se/sv_/Vara-tjanster/bestalla-mikrodata/ (the Statistics Sweden) for detailed information about how to apply for access to register data for research purposes. |

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
