## [Decision Letter]

Thank you for submitting your work entitled "Childhood injury after a parental cancer diagnosis" for peer review at *eLife*. We are pleased to inform you that your submission has been favorably evaluated by Prabhat Jha (Senior editor) and two reviewers, one of whom (Eduardo Franco) is a member of our Board of Reviewing Editors.

The reviewers have discussed the reviews with one another and the Reviewing editor has drafted this decision to help you prepare a revised submission.

The effects of serious parental illness on children's injuries is an interesting topic with potential health policy implications. The study is based on good quality data, and the research questions are clearly motivated.

1) Intuitively, you could have conducted a simpler study, e.g., a case-control study resorting to data linkage to the Swedish Cancer Registry. Instead, you conducted a much stronger cohort study using the opportunities for effective record linkage with administrative and healthcare databases in Sweden. It seems to be the first study of this kind. Please underscore this fact briefly in the Discussion, i.e., the enhanced validity that presumably came from this more robust design.

2) Your contention that the state of cohabitation (unmeasured possible confounder) could be a problem is well reasoned. It is conceivable that there were more divorces around the time of cancer diagnosis. Cancer is devastating to a couple's wellbeing and may lead to a situation that triggers a separation, which in turn may lead to neglect by the caregiving parent in exerting proper surveillance to the child. It is also possible that the divorce will have been the triggering stressful event preceding a cancer diagnosis (a typical anecdote that oncologists like to recall). Either way, the ensuing separation may further contribute to injuries due to the departure of one of the parents from the household. Another scenario favoring injuries is also consistent with the above conditions. Separated or divorced parents may have joint custody of the child, which entails travel between two households and exposure to unfamiliar environments for the child, which in turn could lead to injuries. You should consider the extent of these possible conditions with respect to the social and behavioral mechanisms behind their findings. Please expand the Discussion to elaborate on these possible mechanisms.

3) Although the number of children that are exposed to parental cancer are undoubtedly increasing (one reason being postponement of childbearing), the proportion is still very small. Please acknowledge this fact in the Discussion. Consider also adding a percentage to the number of exposed (Results, first paragraph).

4) The linking of parents and children is not entirely clear. You seem to refer to parents as biological parents of the children. You did not consider whether the children actually reside with their parents (and siblings). This is relevant with regard to the exposure to an ill parent and should be better acknowledged.

5) Expand on the reasons for your choice of covariates in the multivariate models, especially in the Discussion where you noted that most covariates had negligible impact (sixth paragraph). For example, although statistically significant, the differences with regard to birth weight, gestational age and mode of delivery were very small. If the tendency of adjustments was to render the associations stronger, it would be useful to know which covariates made the difference, which would provide insights as to the mechanisms leading to injuries.

6) You state that the records for hospital care had to have a main discharge diagnosis of injury (ICD 10: S00-T98) and an external cause (ICD 10: V01-Y98). Does this mean that hospital care episodes due to complications of medical and surgical care were included? Considering the main motivation for this study, these episodes should have been excluded as the focus lies on the children's proneness for injuries.

7) The follow-up starts on 1.1.2001, however, you stated that children whose parent had an earlier diagnosis contributed all follow-up time to the exposed period (subsection “Follow-up”). How were these children treated in the analyses that used time since cancer diagnosis?

8) Did the risks vary at all by injury severity (subsection “Primary analysis”)? This is an interesting point as hospitalizations are presumably less affected by the parent's propensity to seek care for their children.

9) It is surprising that the risks did not differ by child's age as the typical injuries and their context do differ a lot between toddlers and teenagers. Would a different choice of age groupings have made a difference? For instance, the 12-18 group is quite heterogeneous in terms of dependence from parents. This is also the age group that contributes the vast majority of all events in the follow-up. Splitting this group could have been informative.

10) Make it clear in the Abstract that all the reported hazard ratios are adjusted for important confounders and covariates.

---

## [Author Response]

*1) Intuitively, you could have conducted a simpler study, e.g., a case-control study resorting to data linkage to the Swedish Cancer Registry. Instead, you conducted a much stronger cohort study using the opportunities for effective record linkage with administrative and healthcare databases in Sweden. It seems to be the first study of this kind. Please underscore this fact briefly in the Discussion, i.e., the enhanced validity that presumably came from this more robust design.*

Thank you for the suggestion. We have now stressed this strength in the Discussion section (sixth paragraph).

*2) Your contention that the state of cohabitation (unmeasured possible confounder) could be a problem is well reasoned. It is conceivable that there were more divorces around the time of cancer diagnosis. Cancer is devastating to a couple's wellbeing and may lead to a situation that triggers a separation, which in turn may lead to neglect by the caregiving parent in exerting proper surveillance to the child. It is also possible that the divorce will have been the triggering stressful event preceding a cancer diagnosis (a typical anecdote that oncologists like to recall). Either way, the ensuing separation may further contribute to injuries due to the departure of one of the parents from the household. Another scenario favoring injuries is also consistent with the above conditions. Separated or divorced parents may have joint custody of the child, which entails travel between two households and exposure to unfamiliar environments for the child, which in turn could lead to injuries. You should consider the extent of these possible conditions with respect to the social and behavioral mechanisms behind their findings. Please expand the Discussion to elaborate on these possible mechanisms.*

We agree and have now added comments in this regard in the Discussion section (sixth paragraph).

*3) Although the number of children that are exposed to parental cancer are undoubtedly increasing (one reason being postponement of childbearing), the proportion is still very small. Please acknowledge this fact in the Discussion. Consider also adding a percentage to the number of exposed (Results, first paragraph).*

We agree and have now added comments in this regard in the Discussion section (fourth paragraph). We also as suggested added a percentage to the number of exposed children in the Results section (first paragraph).

*4) The linking of parents and children is not entirely clear. You seem to refer to parents as biological parents of the children. You did not consider whether the children actually reside with their parents (and siblings). This is relevant with regard to the exposure to an ill parent and should be better acknowledged.*

We indeed only studied cancer diagnosis among biological parents of the children. We have now added more information regarding the Swedish Multi-Generation Register and clarified that we only included children with both biological parents identifiable from this register in the present study (subsection “Study participants”).

We also agree that it is important to acknowledge that in the present study we were unable to identify whether or not the child was actually residing with the parents at the time of parental cancer diagnosis. We have added comments in this regard in the Discussion section (sixth paragraph).

*5) Expand on the reasons for your choice of covariates in the multivariate models, especially in the Discussion where you noted that most covariates had negligible impact (sixth paragraph). For example, although statistically significant, the differences with regard to birth weight, gestational age and mode of delivery were very small. If the tendency of adjustments was to render the associations stronger, it would be useful to know which covariates made the difference, which would provide insights as to the mechanisms leading to injuries.*

In the multivariable models, only adjustment for child age at follow-up and parental ages at cancer diagnosis had noticeable impacts on the hazard ratios (HRs) for childhood injuries, whereas adjustment for other covariates had negligible effects. The crude incidence rate ratios for childhood injury any time after parental cancer diagnosis and during the first year after parental cancer diagnosis were 1.12 (95% CI: 1.10-1.13) and 1.30 (95% CI: 1.25-1.35) respectively. For the corresponding HRs only adjusted for child age, adjusted for both child age and parental ages at child birth and the fully adjusted HRs, please refer to the below table.

HR (95% CI) adjusted for child ageHR (95% CI) adjusted for child age and parental ages at child birthFully adjusted HR (95% CI)Any time after parental cancer diagnosis1.04 (1.02-1.06)1.08 (1.06-1.09)1.07 (1.05-1.09)First year after parental cancer diagnosis1.24 (1.19-1.28)1.28 (1.23-1.33)1.27 (1.22-1.33)

We have now further clarified the choice of covariates in the Discussion section (sixth paragraph).

*6) You state that the records for hospital care had to have a main discharge diagnosis of injury (ICD 10: S00-T98) and an external cause (ICD 10: V01-Y98). Does this mean that hospital care episodes due to complications of medical and surgical care were included? Considering the main motivation for this study, these episodes should have been excluded as the focus lies on the children's proneness for injuries.*

Thank you for this very good comment. We have now excluded injuries related to complications of surgical and medical care, i.e. medical injuries defined as injuries with a main discharge diagnosis of ICD 10 codes T80-T88 or T98.3 or with an external cause of ICD 10 codes Y40-Y84 or Y88, from the outcome definition of the analyses. As we expected, among this healthy study population, such injuries comprised a rather small proportion of all injuries identified (among unexposed children, 1.2%, and among exposed children, 1.2%). We have now modified the definition of childhood injury in the Materials and methods section (subsection “Childhood injury”), and updated all results in the Abstract section, Results section (subsections “Primary analysis” and “Secondary analysis”), Figure 1, Table 2, Table 3 and Table 4.

*7) The follow-up starts on 1.1.2001, however, you stated that children whose parent had an earlier diagnosis contributed all follow-up time to the exposed period (subsection “Follow-up”). How were these children treated in the analyses that used time since cancer diagnosis?*

In the analyses taken into account the entire follow-up, children whose parent had a cancer diagnosis before January 1, 2001 contributed all person–time (i.e. person-time experienced from January 1, 2001 to the end of follow-up) to the exposed period. In the analyses where we examined the impact of time since parental cancer diagnosis on childhood injury, we split children’s exposed period into 3 periods, i.e. 1 year or less, >1 and ≤3 years, more than 3 years, after parental cancer diagnosis. For example, if the parent had a cancer diagnosis on January 1, 1999 and the child was followed from January 1, 2001 until December 31, 2010, this child would contribute 1 person-year (January 1, 2001–December 31, 2001) to the “>1 and ≤3 years” category and 9 person-years (January 1, 2002 – December 31, 2010) to the “more than 3 years” category in the analyses. We have now added additional comments in this regard in the Materials and methods section (subsection “Primary analysis”).

*8) Did the risks vary at all by injury severity (subsection “Primary analysis”)? This is an interesting point as hospitalizations are presumably less affected by the parent's propensity to seek care for their children.*

We used the type of hospital contact as a proxy of injury severity, i.e., hospitalization/inpatient visit=more severe, outpatient visit=less severe, and calculated HRs for hospitalization and outpatient visit for injury separately. To compare the HRs of hospitalization and outpatient visit for injury, no direct method was available to our best knowledge. We therefore used a general method for comparing two HRs, exp(β_1_) and exp(β_2_), as previously reported (Altman and Bland, 2003). In brief, we first calculated *z score*= (β_1_– β_2_) / [SE(β_1_)^2^ +SE(β_2_)^2^], then the calculated *z* score was compared to the standard normal distribution, leading to a *P* value. If *P*<0.05, we report that the compared two HRs are statistically significantly different. In the present study, the HR of hospitalization for injury during the entire follow-up was 1.03 (95% CI: 0.99-1.08) and the HR of outpatient visit for injury was 1.08 (95% CI: 1.06-1.10) (section "Results"), leading to a *P* value of 0.10. The HR of hospitalization for injury during the first year after parental cancer diagnosis was 1.18 (95% CI: 1.07-1.31) and the HR of outpatient visit for injury during the first year was 1.29 (95% CI: 1.24-1.35) (section "Results"), leading to a *P* value of 0.20. Although the overall association tended to be stronger for outpatient visit for injury which would support the impact of parent’s healthcare-seeking propensity, the difference was not statistically significant for either any time or the first year after cancer diagnosis. We therefore concluded that we could not draw a conclusion regarding whether or not the HRs for hospitalization and outpatient visit for injury were statistically significantly different from each other. A presumption of this method is that the two compared HRs are estimated from different (independent) samples. In the case of overlapping samples, as in the present study, this formula should be modified by adding a covariance term for β_1_ and β_2_ in the denominator. Although we generally presume that the covariance term is relatively small and has little influence on the estimated *P* values, it is unfortunately rather hard to precisely estimate this covariance term according to our knowledge.

The fact that both hospitalizations and outpatient visits for childhood injuries clearly increased during the first year after parental cancer diagnosis (18% for hospitalization and 29% for outpatient visit) and that the increased risk was limited to the first three years after diagnosis whereas not thereafter, further argued against a pure explanation by differential healthcare-seeking behavior between the affected parents and other parents. We have modified the discussion about this in the Discussion section (sixth paragraph).

*9) It is surprising that the risks did not differ by child's age as the typical injuries and their context do differ a lot between toddlers and teenagers. Would a different choice of age groupings have made a difference? For instance, the 12-18 group is quite heterogeneous in terms of dependence from parents. This is also the age group that contributes the vast majority of all events in the follow-up. Splitting this group could have been informative.*

Thank you for the suggestion. We have further split the 12-18 years group into two groups, i.e. 12-15 years group and 15-18 years group and reported the corresponding HRs with 95% CIs. It seems that there was no evidence that younger and older adolescents had different risks of hospital contacts for injury when a parent had a cancer diagnosis. We have added information in this regard in the Materials and methods section (subsection “Primary analysis”) and presented new results in Table 3.

*10) Make it clear in the Abstract that all the reported hazard ratios are adjusted for important confounders and covariates.*

We have revised the Abstract accordingly.